# Synthesis and H_2_S-Sensing Properties of MOF-Derived Cu-Doped ZnO Nanocages

**DOI:** 10.3390/nano12152579

**Published:** 2022-07-27

**Authors:** Beiying Qi, Xinchang Wang, Xinyue Wang, Jipeng Cheng, Yuanyuan Shang

**Affiliations:** 1Key Laboratory of Material Physics of Ministry of Education, School of Physics and Microelectronics, Zhengzhou University, Zhengzhou 450052, China; qby17516207367@163.com (B.Q.); wxywang98@163.com (X.W.); yuanyuanshang@zzu.edu.cn (Y.S.); 2State Key Laboatory of Silicon Materials, School of Materials Science and Engineering, Zhejiang University, Hangzhou 310027, China; chengjp@zju.edu.cn

**Keywords:** MOF-derived ZnO, nanocages, Cu doping, H_2_S sensing

## Abstract

Metal–organic framework (MOF)-derived pure ZnO and Cu-doped ZnO nanocages were fabricated by calcining a zeolitic imidazole framework (ZIF-8) and Cu-doped ZIF-8. The morphology and crystal structure of the samples were characterized using X-ray diffraction (XRD), scanning electron microscopy (SEM), and high-resolution transmission electron microscopy (HRTEM). It was found that Cu doping did not change the crystal structures and morphologies of MOF-derived ZnO nanocages. The H_2_S-sensing properties of the sensors based on ZnO and Cu-doped ZnO nanocages were investigated. The results indicated that the H_2_S-sensing properties of MOF-derived ZnO nanocages were effectively improved by Cu doping, and the optimal doping content was 3 at%. Moreover, 3 at% Cu-doped ZnO nanocages showed the highest response of 4733 for 5 ppm H_2_S at 200 °C, and the detection limit could be as low as 20 ppb. The gas-sensing mechanism was also discussed.

## 1. Introduction

Hydrogen sulfide (H_2_S) is a colorless, toxic, corrosive, flammable, and acidic gas, and it is produced by natural sources such as the separation of sour gases and the sulfate reducing of hydrocarbons. It can cause damage to the central nervous system for a short period of time, and the leakage of H_2_S gas with levels reaching over 300 ppb in the air imposes tremendous risks on human health [1,2,3,4]. Therefore, the monitoring of H_2_S is very important for environmental protection and human health, and rapid and highly selective detection is the key to H_2_S-gas sensors [5]. In recent years, many types of H_2_S-gas sensors, such as the resistive sensor, the electrochemical sensor, and the surface-acoustic-wave sensor, have been developed. Among these, the resistive sensor has attracted great attention because of its simple fabrication process and excellent sensing performance. Several typical semiconducting metal oxides, including CuO [6,7], ZnO [8,9,10], In_2_O_3_ [11,12], WO_3_ [13], SnO_2_ [14,15], and Co_3_O_4_ [16,17], have been extensively investigated for fabricating gas sensors. However, the gas sensors based on these metal oxides still have several problems, such as low sensitivity, poor selectivity, long response time, and poor detection limits.

Porous metal oxides have highly specific surface areas, and special morphologies and structural characteristics that are conducive to the diffusion and adsorption of the target gas molecules. Recently, gas sensors based on porous metal-oxide materials derived from metal–organic-framework materials have attracted widespread attention due to their unique structural features and good gas sensitivity [18,19]. ZIF-8 is a typical metal–organic-framework structure, and ZIF-8-derived ZnO can retain the morphology, porosity, and highly specific surface area of ZIF-8, which is helpful to improve the gas-sensing properties of ZnO nanomaterials [20,21,22,23]. Xia et al. synthesized well-connected ZnO nanoparticle networks derived by the in situ annealing of ZIF-8, which is more sensitive to ethanol than ex situ-annealed counterparts and commercial ZnO nanoparticles [24]. However, the sensing properties of Cu-doped ZnO nanomaterials derived from Cu-doped ZIF-8 have been rarely investigated.

In this work, we first fabricated ZIF-8 with different Cu-doping content using the homogeneous co-precipitation method and finally obtained a series of Cu-doped ZnO nanocages after calcination at 400 °C using Cu-doped ZIF-8 as the template. The effects of Cu doping on the crystal structure, morphology, and sensing properties of the nanocages were studied in detail.

## 2. Materials and Methods

### 2.1. Materials

C_4_H_6_O_4_Zn·2H_2_O (Aladdin (Shanghai, China), 99%), C_4_H_6_O_4_Cu·H_2_O (Aladdin, 99%), and 2-Methylimidazole (Aladdin, 98%) were purchased from the indicated suppliers and used without any further purification.

### 2.2. Synthesis of ZnO Nanocages and Cu-Doped ZnO Nanocages

In our experiments, 2.94 g of C_4_H_6_O_4_Zn·2H_2_O and 9.84 g of 2-Methylimidazole were dissolved in 40 mL and 60 mL of methanol while stirring for 2 h at room temperature to form uniformly transparent solution A and solution B. Then, solution A was added to solution B drop by drop while the stirring continued. Afterwards, the precursor solutions were stirred for 24 h at room temperature. Finally, the resulting white products were collected by centrifugation and washed with anhydrous ethanol (centrifugal rate of 8000 rpm) for several times to remove the residues; then, they were dried at 60 °C for 12 h under vacuum conditions to obtain ZIF-8. MOF-derived ZnO nanocages were obtained by calcining the collected samples at 400 °C for 2 h in air atmosphere at a heating rate of 2 °C min^−1^. The preparation of Cu-doped ZnO nanocages was similar to that of ZnO nanocages. A certain amount of C_4_H_6_O_4_Cu·H_2_O (1 at%, 3 at%, 5 at%) was added to solution A. The calcination process was the same as that of ZnO nanocages.

### 2.3. Materials’ Characterization

The morphology was characterized using a scanning electron microscope (SEM; JSM-6700F; JEOL, Tokyo, Japan) and a high-resolution transmission electron microscope (HR-TEM; JEOL JEM-2100). The specific surface area and pore-size distribution of the samples were analyzed following nitrogen-adsorption–desorption measurements (NOVA 4200e; Quantachrome, Boynton Beach, FL, USA). The crystalline structure of the samples was characterized with X-ray diffraction (XRD) recorded on a Bruker D8 advanced diffractometer (Billerica, MA, USA).

### 2.4. Fabrication and Analysis of Gas Sensors

An appropriate amount (10 mg) of as-synthesized ZnO nanocages was mixed with deionized water (2 mL) and formed a homogeneous suspension after ultrasonic stirring. ZnO pastes were coated on the surface of gold-plated fork-finger electrodes (ceramic substrate; there were 36 pairs Au fork fingers with an electrode spacing of 0.1 mm) and then placed in a drying oven at 120 °C for 6 h. The gas-sensing properties of the samples were tested using an intelligent gas-sensitive tester (CGS-4TP) produced by Beijing Elite Technology Co., Ltd. (Beijing, China). The response of the sensor was defined as S = Ra/Rg (where Ra and Rg are the sensors’ resistance in the air and in the target gases, respectively). The response time was defined as the time for the response value to reach 90% of its maximum one after the injection of H_2_S, while the recovery time was defined as the time for the response value to decrease to 10% of its maximum one after H_2_S was replaced by air [25].

## 3. Result and Discussion

### 3.1. Morphology and Structure

The specific surface and pore-size distribution of the samples were characterized following N_2_-adsorption–desorption measurements at 77 K (Figure 1). After calculation, the BET surface area of ZIF-8 was 1747.49 m^2^/g, confirming that it had the characteristic of a large specific surface area. Moreover, the nitrogen-sorption isotherm of ZIF-8 was a type-IV adsorption isotherm with an H3-type hysteresis loop at relative pressures of 0.4–1.0, suggesting the existence of both mesopores and macropores. This could also be observed in the pore-size distribution (Figure 1b).

The crystallinity of the as-synthesized samples was examined with XRD analyses. As shown in Figure 2a, all the samples exhibited strong diffraction peaks, and the diffraction peaks in the spectra corresponded to the (110), (200), (211), (220), (310), and (222) reflections of ZIF-8, respectively [26]. No characteristic peaks of other impurities were observed in the XRD patterns. The results indicated that the crystal structure of ZIF-8 could be obtained with different Cu contents and that Cu doping had no obvious influence on the crystal structure of ZIF-8. As shown in Figure 2b, pure ZnO and Cu-doped ZnO nanocages had similar diffraction peaks with good agreement with the hexagonal wurtzite structure of ZnO (PDF#36-1451). For Cu-doped ZnO nanocages, no impure diffraction peaks corresponding to Cu or to the Cu compound were detected, suggesting that Cu ions were efficiently doped into ZnO lattice. This substitution doping may form more surface active sites such as oxygen vacancy on the surface of nanomaterials, which can be conducive to increasing oxygen adsorption on the surface of ZnO nanocages and improving the gas-sensitive properties.

The morphologies and microstructures of the samples were characterized with SEM and TEM. The SEM images of Cu-doped ZIF-8 with different Cu contents (0 at%, 1 at%, 3 at%, 5 at%,) are shown in Figure 3. Figure 3 displays that the samples consisted of uniformly monodispersed polyhedral microparticles. It can be seen from Figure 3 that ZIF-8 and Cu-doped ZIF-8 presented regular polyhedral structures with relatively uniform diameters (250–300 nm). The results showed that a small amount of Cu doping had no obvious effects on the morphology and structure of ZIF-8. Figure 4a–d display the typical SEM images of MOF-derived ZnO and Cu-doped ZnO nanocages after calcination in the air at 400 °C. As can be seen from Figure 4a–d, the surface of ZnO nanocages was relatively rough. Moreover, ZnO hollow nanocages after calcination showed good uniformity (nanocage diameters of 100–150 nm) and retained the original morphological characteristics of ZIF-8. The decrease in the particle diameters was mainly due to the pyrolysis of the organic skeleton, which facilitated the formation of porous structures. These porous structures with large specific surface areas were beneficial to enhance the surface adsorption on ZnO nanomaterials of the target gas and improve their sensing properties. To further investigate the details of the microstructures, 3 at% Cu-doped ZnO nanocages were studied with TEM as a typical example. It can be found from Figure 4c that 3 at% Cu-doped ZnO nanocages had rough surfaces compared with ZnO nanocages. According to the high-magnification TEM image (Figure 4e), a lattice fringe of 0.24 nm was clearly visible and corresponded to the (101) plane of the hexahedral-wurtzite-structure ZnO (PDF#36-1451), indicating that a small amount of Cu doping did not obviously change the crystal structure of ZnO nanocages. To determine the amount of Cu doping in Cu-doped ZnO nanocages, an energy spectrum (EDS) measurement was carried out, and the result is shown in Figure 4f. It can be seen that the sample contained Zn, Cu, and O elements. According to the atomic proportion of Zn and Cu, the doping amount of Cu was 2.7 at% for 3 at% Cu-doped ZnO nanocages, which was relatively close the theoretical value.

### 3.2. Gas-Sensing Properties

In order to investigate the effect of Cu doping on the H_2_S-sensing properties of MOF-derived ZnO nanocages, the H_2_S-sensing properties of all samples were measured systematically. As shown in Figure 5, it was found that the responses of all samples increased with the increase in the operating temperature and then decreased with the further increase in the operating temperature. The optimum operating temperatures of pure ZnO and Cu-doped ZnO nanocages were 225 °C and 200 °C, indicating that Cu doping reduced the optimal operating temperature of MOF-derived ZnO nanocages. Moreover, it can be seen from Figure 5 that the responses of all Cu-doped ZnO nanocages were higher than those of pure ZnO nanocages, suggesting that Cu doping is a feasible way to improve the response of MOF-derived ZnO nanocages. The response of the sensor based on 3 at% Cu-doped ZnO nanocages to 5 ppm H_2_S was 4733, which was much higher than those of pure ZnO nanocages. The responses of 1 at% and 5 at% Cu-doped ZnO nanocages to 5 ppm H_2_S were 1160 and 2600, respectively.

Figure 6a–e shows the response–recovery curves of the samples as functions of H_2_S concentration at the respective optimal operating temperatures. With the increase in H_2_S concentration, the responses of all samples obviously increased. Compared with pure ZnO nanocages, the detection concentration of H_2_S in 3 at% Cu-doped-ZnO-nanocage sensors decreased from 1 ppm to 20 ppb, which indicates that the appropriate amount of Cu doping can reduce the detection limit of H_2_S gas. Figure 6f shows the response/recovery times of all samples. It was found that the response and recovery performances of Cu-doped ZnO nanocages were all better than those of pure ZnO nanocages, and the response/recovery times of 3 at% Cu-ZnO nanocages were 23 s and 53 s for 5 ppm H_2_S at 200 °C. The results showed that Cu doping was beneficial for improving the response/recovery properties of ZnO nanocages. Figure 7 shows the responses of all samples to 5 ppm H_2_S and 100 ppm other interfering gases (NH_3_, CH_4_, NO, NO_2_, Acetone, Methanol, Benzene, Formaldehyde, and Xylene). Compared with pure ZnO nanocages, Cu-doped ZnO nanocages with different Cu contents showed better sensing selectivity. It can be seen from Figure 7 that 3 at% Cu-doped ZnO nanocages showed the highest response to H_2_S, indicating that 3 at% Cu-doped ZnO nanocages exhibited considerable selectivity for H_2_S. Based on the above results, it is reasonable to believe that MOF-derived Cu-doped ZnO nanocages are potentially applicable for detecting H_2_S.

The stability of oxide-semiconductor gas sensors is an important parameter in device applications. Figure 8 shows the stability curves of all samples for 5 ppm H_2_S at their respective optimum operating temperatures. It can be seen that both Cu-doped ZnO nanocages with different Cu contents and pure ZnO nanocages showed good stability.

### 3.3. Gas-Sensing Mechanism

The sensing mechanism of gas sensors based on metal-oxide semiconductor belongs to the “redox” process, in which gas molecules adsorb and desorb on the surface of the sensor, thus leading to the change in resistance [27,28,29]. When a sensor based on ZnO nanocages is exposed to the air, oxygen molecules can easily adsorb on the surfaces of the nanostructures, capture free electrons, and exist as chemisorbed oxygen species (O_2_^−^, O^−^, O^2^^−^) [30,31]; then, the resistance of the sensor increases. When exposed to H_2_S, H_2_S molecules may react with the chemisorbed oxygen ions and release electrons to ZnO nanomaterials. As a result, the resistance of the sensor eventually decreases. It can be expressed as the following reaction [5,30,32]:O_2_ (ads)+ 2e^−^ →2O^−^ (ads)    (100–300 °C)(1)
H_2_S (g) + 3O^−^ (ads)→ H_2_O (g) + SO_2_ (g) + 3e^−^(2)

Derived ZnO nanocages calcined by ZIF-8 have the porous structure of ZIF-8, which enhances the adsorption on ZnO nanocages of oxygen and the target gas and improves the H_2_S-sensing properties of ZnO nanocages. Cu doping can introduce more oxygen-vacancy defects on the surface of ZnO nanocages, thus facilitating the adsorption of more oxygen species on the surface of Cu-doped ZnO nanocages. In addition, the surface oxygen vacancy serves as a place for gas-sensing reactions. The increase in oxygen vacancies on the nanocage surface is beneficial to promote the gas-sensing reaction of oxygen species and H_2_S molecules, which can significantly improve the gas-sensing properties of ZnO nanocages. Moreover, CuO is a typical basic metal oxide, and these alkaline-sensitive material surfaces are more beneficial for conducting gas-sensing reactions and forming highly conductive CuS when the sensors based on Cu-doped ZnO nanocages are exposed to H_2_S [33,34]. Thus, the H_2_S sensing properties of Cu-doped ZnO nanocages are improved.

## 4. Conclusions

In summary, ZIF-8 and Cu-doped ZIF-8 were synthesized at room temperature using methanol as solvent; then, the derived pure ZnO and Cu-doped ZnO nanocages were prepared by calcining ZIF-8 and Cu-doped ZIF-8. MOF-derived pure ZnO and Cu-doped ZnO nanocages retained the porous structure and unique morphology of ZIF-8, and Cu doping did not change the crystal structures and morphology of ZIF-8 and MOF-derived ZnO nanocages. The H_2_S-sensing measurements showed that the sensing properties to MOF-derived ZnO nanocages were greatly enhanced by Cu doping, and the optimal doping concentration was 3 at%. Compared with undoped ZnO nanocages, 3 at% Cu-doped ZnO nanocages showed the highest response of 4733 for 5 ppm H_2_S at 200 °C.

## Figures and Tables

**Figure 1 nanomaterials-12-02579-f001:**
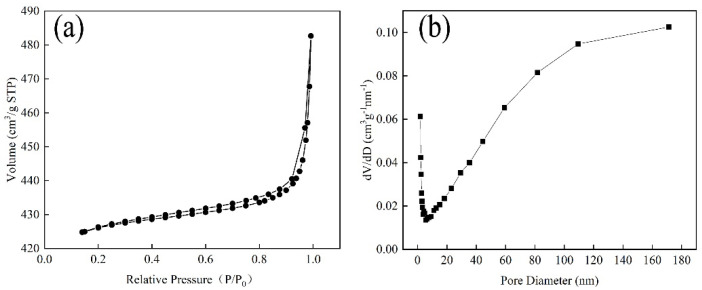
(**a**) Adsorption/desorption curves of ZIF-8 and (**b**) pore-size distribution of ZIF-8.

**Figure 2 nanomaterials-12-02579-f002:**
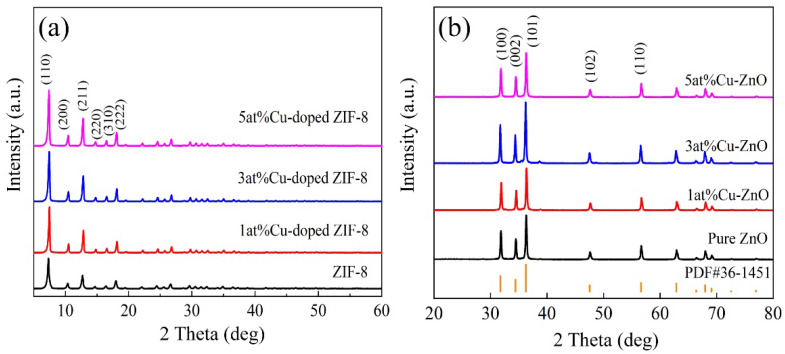
XRD patterns of the samples: (**a**) ZIF-8 and Cu-doped ZIF-8; (**b**) MOF-derived ZnO and Cu-doped ZnO nanocages.

**Figure 3 nanomaterials-12-02579-f003:**
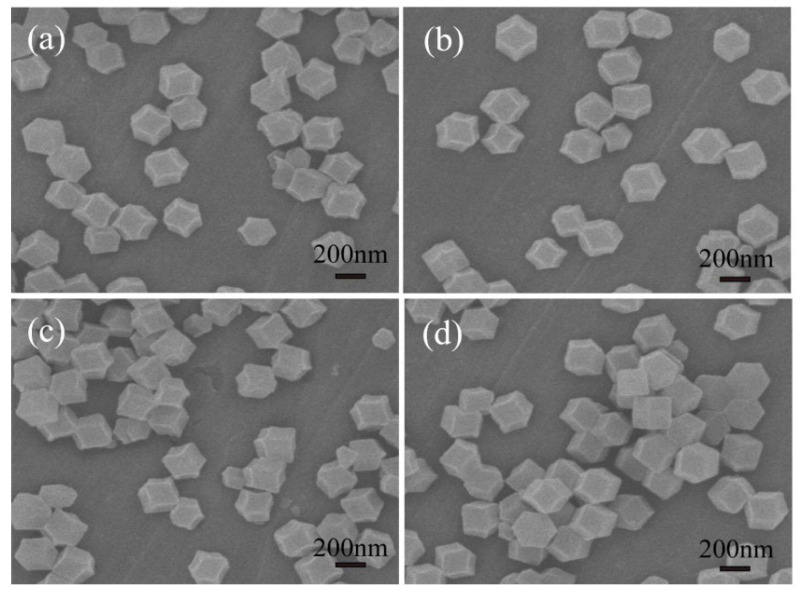
SEM images of (**a**) ZIF-8 and (**b**–**d**) Cu-doped ZIF-8 with different Cu contents (1 at%, 3 at%, 5 at%).

**Figure 4 nanomaterials-12-02579-f004:**
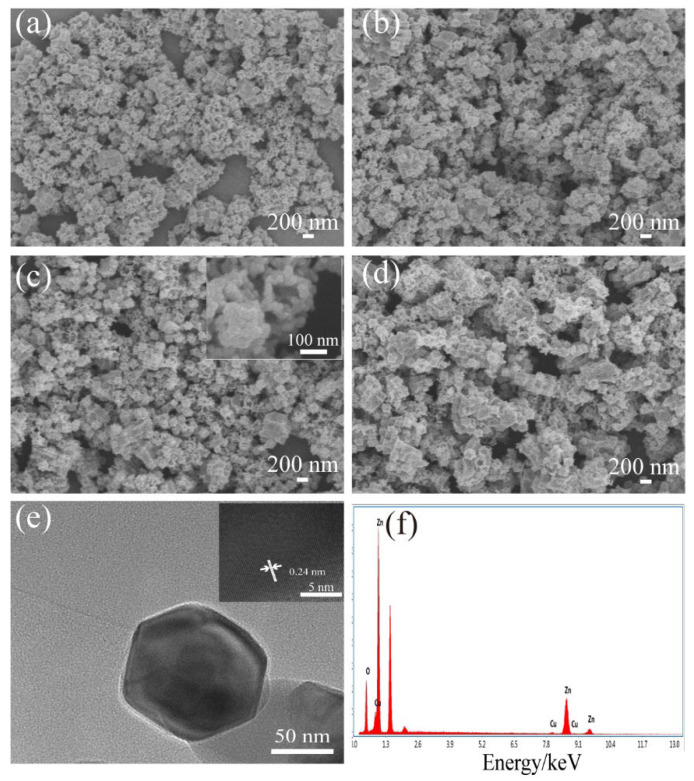
SEM images of the samples: (**a**) pure ZnO nanocages, (**b**–**d**) Cu-doped ZnO nanocages with different Cu contents (1 at%, 3 at%, 5 at%), (**e**) TEM image, the inset in (**e**) is an HRTEM image and (**f**) EDS spectra of 3 at% Cu-doped ZnO nanocages.

**Figure 5 nanomaterials-12-02579-f005:**
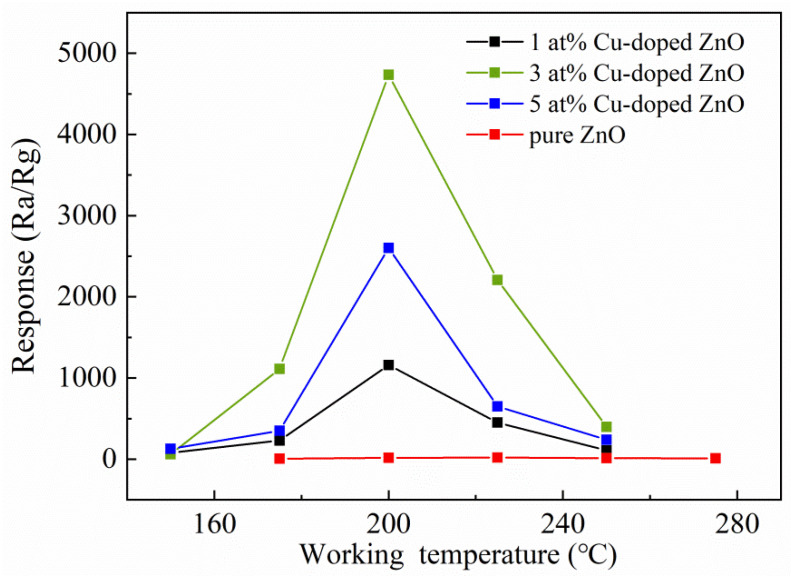
Response curves of ZnO nanocages with different Cu contents with respect to H_2_S at different operating temperatures.

**Figure 6 nanomaterials-12-02579-f006:**
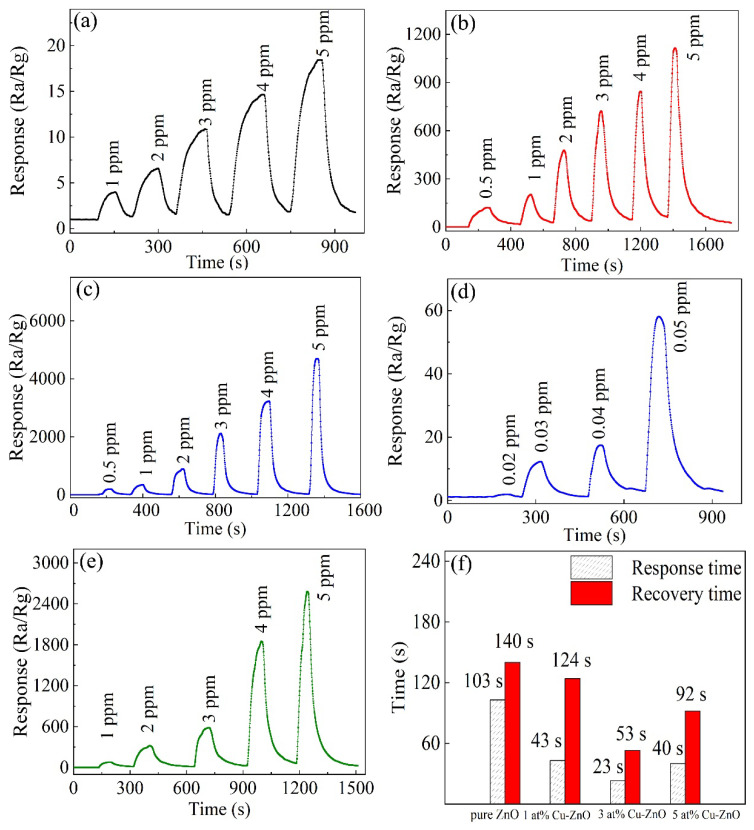
Dynamic-response curves of the sensors based on (**a**) pure ZnO nanocages, (**b**) 1 at% Cu-doped ZnO nanocages, (**c**,**d**) 3 at% Cu-doped ZnO nanocages, and (**e**) 5 at% Cu-doped ZnO nanocages for different concentrations of H_2_S at their optimal operating temperatures. (**f**) Response and recovery times of all samples.

**Figure 7 nanomaterials-12-02579-f007:**
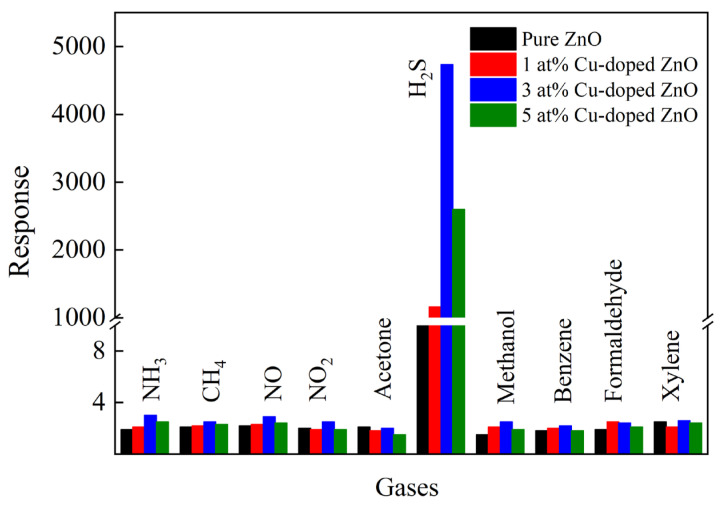
The selectivity of all samples for different gases.

**Figure 8 nanomaterials-12-02579-f008:**
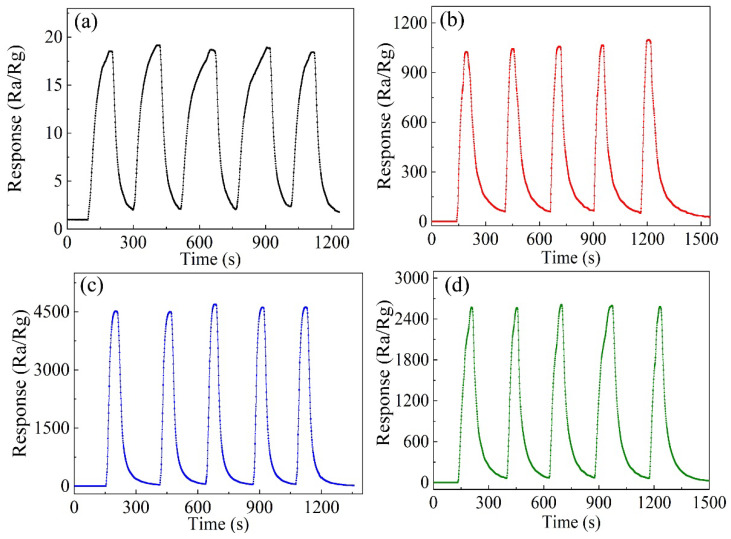
The stability of different samples for 5 ppm H_2_S at their optimum operating temperatures: (**a**) pure ZnO nanocages, (**b**) 1 at% Cu-doped ZnO nanocages, (**c**) 3 at% Cu-doped ZnO nanocages, and (**d**) 5 at% Cu-doped ZnO nanocages.

## Data Availability

The data presented in this study are available on request from the corresponding author.

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
