# Peer review of "Synthesis and H_2_S-Sensing Properties of MOF-Derived Cu-Doped ZnO Nanocages"

_nanomaterials, 2022, doi:10.3390/nano12152579_

Round 1

Reviewer 1 Report

All abbreviations used should be explained at the beginning, preferably in Abstract. In particular it concerns “MOF, ZIF-8". 

Grammatical and orthographic errors and remarks concerning the figures are listed below. 

Line 35: what means “sensral” 

Lines 46,47: grammatical errors 

Line 57: grammatical error, it should be a sentence 

Line 81: what means “enhased” 

Line 104: grammatical error 

Figure 4: horizontal axis should be labeled as “Temperature (o C)” 

Figures 5(a)-(d): too small magnification, no details are seen 

Lines 196, 197, 203, 220: orthographic errors. 

Author Response

Response to the reviewers’ comments

The authors thank reviewers for your comments and suggestions. These comments are all valuable and very helpful for revising and improving our work, as well as the important guiding significance to our research. We have gone through the reviewers’ comments carefully and have made revisions. The responses to the reviewers’ comments point by point are presented as below.

Reviewer 1

Comments and Suggestions for Authors

  1. All abbreviations used should be explained at the beginning, preferably in Abstract. In particular it concerns “MOF, ZIF-8".

Response: Thank you for your advice. We have explained the abbreviations of MOFs and ZIF-8 in abstract.

  1. Grammatical and orthographic errors and remarks concerning the figures are listed below.

Line 35: what means “sensral”

Lines 46,47: grammatical errors

Line 57: grammatical error, it should be a sentence

Line 81: what means “enhased”

Line 104: grammatical error

Figure 4: horizontal axis should be labeled as “Temperature (o C)”

Figures 5(a)-(d): too small magnification, no details are seen

Lines 196, 197, 203, 220: orthographic errors.

Response: Thank you for your advice for pointing out some problems in our manuscript. We have marked the revised content in red font in the text.

Reviewer 2 Report

Dear Authors, 

I have some comments and suggestions that will be strengthen this work. 

-          Introduction requires improvements and incorporation of the following:

1)      Some information/background on H2S ie where its found and formed from and abundancy 

2)      What are acceptable levels of H2S for human health and the environmental

3)      A potential application the sensor could be used in as the operating temperature is very specific (ie 200oC)

4)      Include introduction on why Cu doping will improve sensor performance or it’s importance

-          Materials and Method requires the following:

1)      Please include details of heating ramp rate and atmosphere conditions for the synthesis of ZnO nanocages

2)      Please include gas adsorption characterisation of ZIF-8 to determine porosity (ie surface area, pore size distribution, N2@77K).

3)      “An appropriate amount” is not acceptable for fabrication details. This includes the volume of deionized water. Please include all fabrication details.

4)      Thickness of coating should be provided. Please provide details. 

5)      Please include details of your gas sensing conditions (ie carrier gas details, H2S concentration range of operation)

-          Results and Discussion

1)      Please include details of H2S concentration range and carrier gas details in main text.

2)      Please discuss the advantage of sensing at 200oC instead of 225oC and why room temperature experiments were not reported or performed

3)      Porous metal oxides were introduced but yet you make no mentions or  attempt to discuss the porous nature of the ZnO nanocages or the Cu doped systems. No discussion on the benefits of the nanocages or relevance to improving H2S sensing. This should be discussed. 

4)      Figure 4 – X-axis is labelled incorrectly (should read Temp oC not time (s)

5)      Why are there no room temperature experiments? 

6)      Figure 6 – define operating temperature in caption for the reader 

7)      Labels in Figure 5 a-d need to be clearer

8)      It is not clear at all where a 20ppb detection limit was determined. Can this please be explained and supported by data?

9)      Ideally this paper should have H2S in different ranges of concentration, not only 5 to 1ppm.

10)   Typo page 3 line 95 “cxamined” should read “examined”

11)   Typo page 7 line 197 “recat” should read “react”

12)   Clarify – Figure 5 is the response of all Cu-ZnO nanocages performed in the presences of the interfering gases? If not, perhaps an experiment should be done where by H2S is in the presences of an interfering gas. This would show the great selectivity of H2S in the presence of other gases with different physical and chemical properties.

13)    The value of Cu doping is not provide till the end and should be noted earlier to have the reader understand the relevance of doping to this body of work.

14) dive into the benefits of high temp detections or the limitations of it for sensors applications. 

Kind regards. 

Author Response

Response to the reviewers’ comments

The authors thank reviewers for your comments and suggestions. These comments are all valuable and very helpful for revising and improving our work, as well as the important guiding significance to our research. We have gone through the reviewers’ comments carefully and have made revisions. The responses to the reviewers’ comments point by point are presented as below.

Reviewer 2

I have some comments and suggestions that will be strengthen this work. 

-          Introduction requires improvements and incorporation of the following:

1) Some information/background on H2S ie where its found and formed from and abundancy 

Response: Thank you for your advice. We have marked the revised content in red font in the text.

2) What are acceptable levels of H2S for human health and the environmental

Response: Thank you for your advice. We have marked the revised content in red font in the text.

3) A potential application the sensor could be used in as the operating temperature is very specific (ie 200oC)

Response: Compared with the operating temperature of the common oxide semiconductor sensors (above 300 oC), the operating temperature of Cu-doped ZnO nanocage sensors is significantly reduced, which will help to reduce the power consumption of gas sensor.

4) Include introduction on why Cu doping will improve sensor performance or it’s importance

Response: As an important part of this paper, the analysis of Cu doping is put into the gas-sensing mechanism part for the overall analysis.

-          Materials and Method requires the following:

1) Please include details of heating ramp rate and atmosphere conditions for the synthesis of ZnO nanocages.

Response: Thank you for your advice. We have added the calcined parameters and marked the revised content in red font in the text.

2) Please include gas adsorption characterisation of ZIF-8 to determine porosity (ie surface area, pore size distribution, N2@77K).

Response: According to your comments, we added to the BET results of ZIF-8 (Figure 1) and the analysis sentences as follows: ZIF-8 has the characteristics of large specific surface area and porous, so we test its specific surface area. Figure 1(a) shows the N2 adsorption/desorption curves of ZIF-8. After calculation, the specific surface area of ZIF-8 is 1747.49 m2/g, confirming that it has the characteristic of large specific surface area. Figure 1(b) shows the pore size distribution of ZIF-8. The material contains a large number of micropores, most of which are distributed at about 10 nm.

3) “An appropriate amount” is not acceptable for fabrication details. This includes the volume of deionized water. Please include all fabrication details.

Response: Thank you for your advice. We have added the required data and marked the revised content in red font in the text.

4) Thickness of coating should be provided. Please provide details. 

Response: ZnO nanocages (10 mg) were dispersed into deionized water (2 ml) to form a homogeneous suspension, then dropped onto the fork finger electrode. The coating thickness was very thin near a monolayer.

5) Please include details of your gas sensing conditions (ie carrier gas details, H2S concentration range of operation).

 Response: In this experiment, the static method was used to test the gas-sensing properties of the gas sensor based on ZnO nanocages, and the required amount of H2S was injected into the gas chamber. Therefore, the carrier gas was not used in this experiment. The H2S test concentration was above 0.01 ppm.

-          Results and Discussion

1) Please include details of H2S concentration range and carrier gas details in main text.

Response: The H2S test concentration was above 0.01 ppm, and the carrier gas was not used in this experiment.

2)  Please discuss the advantage of sensing at 200oC instead of 225oC and why room temperature experiments were not reported or performed

Response: The sensor based on ZnO nanocages with different Cu contents does not show H2S sensing performances (Ra/Rg=1) at room temperature and low temperature, so it cannot be reported at the room temperature. Moreover, We found that the operating temperature of the sensor based on ZnO nanocages can be appropriately reduced by Cu doping, which would be benefit to reduce the energy consumption of the gas sensor.

3) Porous metal oxides were introduced but yet you make no mentions or  attempt to discuss the porous nature of the ZnO nanocages or the Cu doped systems. No discussion on the benefits of the nanocages or relevance to improving H2S sensing. This should be discussed. 

Response: According to your comments, we added to the sentences as follows: Derived ZnO nanocages calcined by ZIF-8 have the porous structure of ZIF-8, which will be benefit to enhance the adsorption of ZnO nanocages to oxygen and the target gas, and improve H2S sensing properties of ZnO nanocages.

4) Figure 4 – X-axis is labelled incorrectly (should read Temp oC not time (s)

Response: According to your comments, we have modified Figure 4.

5) Why are there no room temperature experiments? 

Response: The sensor based on ZnO nanocages with different Cu contents does not show H2S sensing performances (Ra/Rg=1) at room temperature and low temperature, so it cannot be reported at the room temperature.

6) Figure 6 – define operating temperature in caption for the reader 

Response: Thank you for your suggestion to improve our manuscript. The operating temperature of gas sensor is the temperature of the sensor at the optimal gas-sensing performance. This parameter is a well-known parameter in the field of gas sensors, and we do not think it is necessary to specifically define the operating temperature.

7) Labels in Figure 5 a-d need to be clearer

Response: According to your comments, we have modified Figure 5.

8) It is not clear at all where a 20ppb detection limit was determined. Can this please be explained and supported by data?

Response: Thank you for your suggestion to improve our manuscript. The corresponding curve for low concentrations of H2S (20 ppb-50 ppb) have been placed in Figure 6(d).

9) Ideally this paper should have H2S in different ranges of concentration, not only 5 to 1ppm.

Response: Thank you for your suggestion to improve our manuscript. Since H2S is a highly toxic gas, it usually needs to be detected at a very low concentration. Thus the test concentration of H2S in this paper is set at from 0.02ppm to 5ppm.

10) Typo page 3 line 95 “cxamined” should read “examined”

Response: Thank you for your advice for pointing out the problem in our manuscript. We have marked the revised content in red font in the text.

11) Typo page 7 line 197 “recat” should read “react”

Response: Thank you for your advice for pointing out the problem in our manuscript. We have marked the revised content in red font in the text.

12) Clarify – Figure 5 is the response of all Cu-ZnO nanocages performed in the presences of the interfering gases? If not, perhaps an experiment should be done where by H2S is in the presences of an interfering gas. This would show the great selectivity of H2S in the presence of other gases with different physical and chemical properties.

Response: Thank you for your suggestion to improve our manuscript. We have divided Figure 5 into Figures 6 and Figure 7. Figure 6 show the H2S sensing properties of ZnO nanocages with different Cu content in the air atmosphere, while Figure 7 show the responses of the interference gases in the air atmosphere under the same test conditions. Then these results were contrasted with the response of H2S. This test method is the same as the test method mentioned by the reviewer.

13) The value of Cu doping is not provide till the end and should be noted earlier to have the reader understand the relevance of doping to this body of work.

Response: Thank you for your suggestion to improve our manuscript. The value of Cu doping is not provide at the sention of materials and methods.

14) dive into the benefits of high temp detections or the limitations of it for sensors applications. 

Response: The operating temperature of oxide semiconductor gas sensors typically usually need to reach 300-400 oC, and this high operating temperature of oxide semiconductor gas sensor is one of its disadvantages. At present, the research focus in this field is on reducing the operating temperature, which can reduce the power consumption of oxide semiconductor gas sensor and broaden its application scope.

Reviewer 3 Report

This is a nice piece of work on the MOF bases sensor material. The authors say on pages 3 and 7 that the improved sensing may be due to the oxygen vacancies generated bu Cu ions. I think that may be true, if there are Cu(I)ions on the surface. Is there any information on the oxidation state of Cu or is it just speculation? As far as I understand, the EDS spectra do not give such data.

Author Response

Response to the reviewers’ comments

The authors thank reviewers for your comments and suggestions. These comments are all valuable and very helpful for revising and improving our work, as well as the important guiding significance to our research. We have gone through the reviewers’ comments carefully and have made revisions. The responses to the reviewers’ comments point by point are presented as below.

Reviewer 3

Comments and Suggestions for Authors

This is a nice piece of work on the MOF bases sensor material. The authors say on pages 3 and 7 that the improved sensing may be due to the oxygen vacancies generated bu Cu ions. I think that may be true, if there are Cu(I) ions on the surface. Is there any information on the oxidation state of Cu or is it just speculation? As far as I understand, the EDS spectra do not give such data.

Response: We fully agree with your opinion. In this study, Cu doping is beneficial to significantly improve H2S sensing properties of ZIF-8 derived ZnO nanocages, and it can be speculated that Cu ions may exist in ZnO nanocages at a monovalent value. Moreover, Cu doping can also introduce more oxygen-vacancy defects on the surface of ZnO nanocages. Due to the campus epidemic control, XPS can not be used to test and analyze the oxidation state of Cu ions, thus the valence of Cu is not available.